# Mentor-Mind: Risk-Aware, Constraint-Grounded Advice Agents Beyond Chain-of-Thought

**GPT-5 Pro**

**Yun Wing Kiang**
Department of Electrical and Electronic Engineering
The University of Hong Kong
Hong Kong
`kiangyw@eee.hku.hk`

## Abstract

Large language models (LLMs) have shown remarkable reasoning abilities through prompting techniques like chain-of-thought (CoT) prompting and self-consistency decoding, achieving state-of-the-art results on complex tasks. However, these methods rely on the model's generated rationales, which can be unreliable – often hallucinating plausible-sounding but unfaithful content – and do not account for risk or hard constraints in decision making. We propose Mentor-Mind, an influence-diagram (ID)–grounded advice agent that marries LLM reasoning with decision-theoretic planning. Mentor-Mind uses domain- and mentor-specific decision graphs (IDs) as structured scaffolds for reasoning, ensuring that recommendations satisfy hard domain constraints and optimize a risk-sensitive objective (e.g. Conditional Value-at-Risk). In synthetic yet complex advisory scenarios (energy facility siting, code review, early-career planning), Mentor-Mind generates advice that is more aligned, faithful, and risk-aware than baseline prompting methods. Experimental results show that our approach maintains high decision quality under uncertainty while strictly respecting constraints, outperforming CoT and self-consistency prompts in both success rate and adherence to safety constraints. This work demonstrates a practical integration of LLMs with symbolic decision frameworks, yielding advice agents that replace the "make-up" CoT reasoning with grounded, trustworthy decision analysis.

## 1  Introduction

Large language models (LLMs) have rapidly advanced in multi-step reasoning abilities. Prompting techniques like chain-of-thought (CoT) prompting guide models to break down problems into intermediate steps, greatly improving performance on arithmetic, commonsense, and symbolic reasoning tasks *(Wei et al., 2022)*. For example, few-shot CoT prompting enables a 540B model to achieve state-of-the-art results on math word problems, and even zero-shot prompts (e.g. appending "Let's think step by step") can trigger impressive reasoning *(Kojima et al., 2022)*. Building on CoT, decoding strategies such as self-consistency sample multiple reasoning paths and select the answer most supported by the majority *(Wang et al., 2023)*, yielding significant accuracy gains. These prompting-based approaches have become de facto baselines for eliciting reasoning in LLMs.

However, fundamental challenges remain when using LLMs for decision support and advice. CoT prompts induce the model to generate reasoning, but do not guarantee the reasoning is correct or faithful to facts. LLM-generated rationales are prone to hallucinations – producing plausible-sounding but untrue or nonsensical content *(Huang et al., 2025)*. This is especially problematic in advisory scenarios, where a confident but incorrect justification can mislead users. Alignment techniques like reinforcement learning from human feedback have improved the general helpfulness and harmlessness of LLM assistants *(Ouyang et al., 2022)*, but they do not fully ensure domain-specific decision quality, respect for hard constraints, or risk-awareness in high-stakes domains. In critical decision-making

settings (finance, medicine, engineering), an LLM advisor must not only avoid toxic or blatantly false outputs but also internalize the preferences, trade-offs, and constraints that a human expert would consider.

Crucially, standard CoT prompting lacks a mechanism to enforce hard constraints and preferences. By grounding the LLM's reasoning in an influence diagram (ID), Mentor-Mind ensures every piece of advice corresponds to a well-defined path in a mentor's decision graph. Rather than having the model make up a chain-of-thought, we supply it with a domain-specific decision graph (elicited from a human mentor or expert). The LLM then "plays the role" of a transparent reasoning engine that traverses this graph, evaluating different decision paths, outcomes, and utilities. In essence, Mentor-Mind marries prompt-driven LLM reasoning with model-based planning: ID acts as a scaffold that constrains the LLM's generation to valid, feasible trajectories.

We evaluate Mentor-Mind in several synthetic yet realistic advisory scenarios. Although the scenarios are simulated, they are designed to mirror real-world decision problems in their respective fields, allowing us to fully control conditions and obtain an oracle solution for evaluation. For example, the energy siting task reflects challenges in power infrastructure planning (balancing profit versus environmental risk), and the secure code review scenario parallels real software security auditing, but with known ground-truth outcomes.

We compare Mentor-Mind against baseline prompting strategies: standard CoT prompting, CoT with self-consistency voting, and a textual "memo" baseline where the model is given the constraints and guidelines in plain language (simulating a user reminder to follow rules). We find that Mentor-Mind consistently produces more reliable and safe recommendations.

## 2 Related Work

**LLM Reasoning via Prompting:** There is a rich body of work on improving LLM reasoning using prompt-based techniques. Chain-of-thought prompting (CoT) has emerged as a simple yet powerful method to induce step-by-step reasoning in large models *(Wei et al., 2022)*. By providing examples of reasoning chains, CoT enables models to solve arithmetic and logical problems that stumped them with direct prompting. Subsequent research showed that even without few-shot examples, adding certain trigger phrases can elicit reasoning: Kojima et al. *(2022)* demonstrated that a zero-shot prompt "Let's think step by step" unlocks surprising reasoning ability. Beyond prompting alone, researchers have explored multi-step inference with LLMs, such as generating explicit scratchpads or logic traces *(Nye et al., 2021)* and using least-to-most prompting *(Zhou et al., 2022)* to recursively break down reasoning tasks.

Another line of work augments LLM reasoning with external tools to improve accuracy and grounding – for example, the ReAct framework *(Yao et al., 2023)* interleaves reasoning and acting by allowing the model to query external APIs or knowledge bases during its chain-of-thought. ReAct demonstrated that by retrieving factual information (e.g. via a Wikipedia API) when needed, LLMs can overcome some hallucinations and produce more correct, interpretable reasoning traces *(Yao et al., 2023)*. Similarly, Toolformer *(Schick et al., 2023)* teaches an LM to call tools like calculators or search engines mid-generation, significantly improving performance on tasks requiring external computation or up-to-date knowledge. Our approach differs in that the "tool" we integrate is a domain-specific decision simulator rather than a general knowledge source.

**Decision-Theoretic Planning:** influence diagrams (IDs) and related decision models have long been used for complex decision-making. *Howard & Matheson (1984)* introduced IDs as an extension of Bayesian networks for representing decision problems. They consist of decision nodes (choices to be made), chance nodes (uncertain factors), and utility nodes (objectives to maximize). Solving an ID via dynamic programming yields an optimal policy that maximizes expected utility *(Nease & Owens, 1997)*. Simulation-based solvers (forward Monte Carlo) for IDs have also been studied *(Charnes Shenoy, 1997)*. Prior work has explored using IDs and graphical models in AI assistants; for example, *Moore & Agogino (1987)* described an expert system that guides knowledge acquisition through an ID structure. However, these approaches typically required manually encoding expert knowledge and did not involve LLMs.

**Hallucinations, Alignment, and Faithfulness:** A well-known limitation of LLM-generated explanations is the tendency to hallucinate – generating details that are not grounded in reality or the

provided context *(Ji et al., 2023)*. Hallucinations undermine the trustworthiness of advice, especially in domains like law or medicine (where confidently stated falsehoods can be dangerous). By constraining the LLM's reasoning within a structured decision graph, Mentor-Mind addresses one aspect of hallucination: it forces the rationale to follow the relationships defined in the ID. The graph explicitly represents uncertainty and uses risk-aware criteria, so it is less likely to give overconfident advice – effectively addressing a form of hallucination where the model is overly optimistic about uncertain outcomes. On the broader alignment front, techniques like Reinforcement Learning from Human Feedback (RLHF) have produced more helpful and harmless LLMs *(Ouyang et al., 2022)*. Our work is complementary: rather than aligning to general human preferences, we ensure mentor-specific alignment by enforcing that the LLM's advice matches a particular mentor's priorities and constraints as encoded in the ID.

## 3 Methodology

### 3.1 Influence Diagram Advisor Framework

Mentor-Mind models the advisory task as an influence diagram (ID) – a directed acyclic graph with decision nodes, chance nodes, and a utility node capturing the mentor's preference model for outcomes. The ID structure defines the information flow and dependencies: arcs into decision nodes indicate what the assistant knows when making each decision (e.g. a decision node might have incoming arcs from previous chance nodes or decisions, meaning those outcomes are observed before the decision). Arcs into chance nodes denote causal or conditional influences (e.g. a chance node might depend on a previous decision). Arcs into the utility node indicate which variables affect the mentor's utility. In our formulation, the assistant's goal is to recommend decisions that maximize the mentor's expected utility (or a risk-sensitive objective) given the information available at each point.

Figure 1 in Appendix A.2 illustrates a simplified ID. Oval nodes represent chance variables (uncertain factors outside the assistant's control, e.g. user reactions, environmental events). The rectangular node represents a decision to be made by the assistant, and the diamond node represents the utility (mentor's payoff or value) which all decisions aim to maximize. The optimal policy under a traditional expected utility criterion can be found by backward induction on the diagram *(Howard & Matheson, 1984)*, but instead of solving it analytically, Mentor-Mind uses the LLM to emulate the decision reasoning process.

Given an ID for a domain, we translate it into a structured prompt for the LLM. This prompt describes the decisions to be made, the relevant uncertainties, and how the mentor values different outcomes. The LLM is then asked to simulate the mentor's reasoning: it considers each possible action, "imagines" what might happen (by stepping through chance nodes), and evaluates the outcome according to the utility function. Crucially, the LLM is constrained to follow the graph – it cannot introduce extraneous factors not in the diagram. The ID serves as a structured scaffold for reasoning, ensuring that advice remains grounded in the specified factors and relationships.

### 3.2 Hard Constraints via Feasibility Filters

A key innovation in Mentor-Mind is the integration of hard constraints directly into the decision-making process. We represent each constraint as a binary feasibility test on potential decision paths. For example, in the energy siting domain, a hard constraint might be "the plan must comply with environmental regulation X" – if a candidate site would violate X, any decision recommending that site is marked infeasible. During the LLM's traversal of the ID, we enforce that it skips over any branch that fails a feasibility test. We enforce feasibility in *two stages*. First, a **programmatic whitelist** removes any infeasible actions *before* the LLM sees the options (action-conditioned checks compiled from the public graph spec). Second, a **post-decision validator** checks the chosen action and, upon violation, triggers a single *re-ask restricted to the feasible set*. Prompts mirror these rules for transparency; enforcement occurs at the I/O boundary, preventing jailbreak/forgetting. See Algorithm 2 in Appendix C for the runtime gate that pre-filters the action set and validates the final choice programmatically. This mechanism guarantees that Mentor-Mind's advice never *knowingly* violates a hard rule. It contrasts with unconstrained CoT prompting which might overlook or forget constraints; here the ID's structure explicitly rules out forbidden decision paths from the outset. By encoding inviolate domain rules as feasibility filters, we achieve *constraint-grounded* reasoning – the LLM's exploration of options is bounded by what is permissible.

### 3.3 Mentor-Specific Utility Modeling

In Mentor-Mind, each domain is paired with a specific mentor profile that defines the utility function. We adopt a multi-attribute utility model: for a given outcome, we compute several attribute scores (e.g. economic profit, environmental impact, user satisfaction) and then take a weighted sum $U(x) = \sum_i w_i\, u_i(x)$, where $u_i(x)$ is the normalized utility for attribute $i$ and $w_i$ is the weight reflecting the mentor's relative preference for that attribute. For instance, a risk-averse environmental mentor might assign a very high weight to environmental impact and lower weight to cost, whereas a profit-driven mentor would do the opposite. By adjusting $w$, we effectively tune the system to different mentor personas. These weights and sub-utility functions are elicited from the mentor (or chosen in synthetic domains to simulate a particular value system). Appendix B details the utility functions and weight choices used in our experiments.

Given this mentor-specific utility model, Mentor-Mind's advice aims to maximize the mentor's expected utility. Importantly, because the ID encodes how decisions and chance events lead to outcomes, we can evaluate the *expected utility* of any policy under the mentor's values. This provides an oracle benchmark – the policy (sequence of decisions) that maximizes the mentor's expected utility represents the gold-standard advice to which we compare the LLM's recommendations. We refer to this optimal policy as the oracle mentor policy or simply "oracle." An ideal LLM advisor would match the oracle's decisions in every scenario.

### 3.4 Risk-Sensitive Decision Objectives

Beyond a standard expected-utility objective, Mentor-Mind introduces risk-sensitive criteria so that the advisor can account for outcome variability and worst-case scenarios. In addition to the default Expected Utility (EU) mode (risk-neutral, maximizing $E[U]$), we implement a Conditional Value-at-Risk criterion – Mentor-Mind can be instructed to maximize $CVaR_\alpha$ [1], the expected utility of the worst $(1-\alpha)$ fraction of outcomes *(Rockafellar & Uryasev, 1999)*. Intuitively, CVaR focuses on the lower tail: it measures performance in the worst-case outcomes. We optimize CVaR with $\alpha$=0.10 (tail mass), i.e., the expected utility in the worst $10\%$ of outcomes; ablations sweep $\alpha \in \{0.05, 0.10, 0.14\}$.[2] This is useful for risk-averse mentors who would rather sacrifice some average reward in exchange for insurance against disaster.

We also allow a weighted Mean–CVaR trade-off: the mentor specifies a mixing weight $\lambda \in [0,1]$. The advisor maximizes the convex combination

$$(1-\lambda)\, E[U] + \lambda\, \text{CVaR}_\alpha[U].$$

Varying $\lambda$ interpolates between risk-neutral ($\lambda$=0) and fully risk-averse ($\lambda$=1) behavior; intermediate values provide a balance. Throughout, we use $\alpha$ as *tail mass* (e.g., $\alpha$=0.10 means the worst $10\%$ tail), and we ablate $\lambda \in \{0, 0.5, 1\}$ and $\alpha \in \{0.05, 0.10, 0.14\}$.

### 3.5 Sampling-Based Utility Estimation

Exactly computing the expected utility or CVaR of a given decision policy can be intractable in complex real-world domains, so Mentor-Mind uses a sampling-based Monte Carlo approach to evaluate candidate decisions. When the advisor is at a decision node, the system simulates many rollouts of the ID for each possible action *(Charnes & Shenoy, 1997)*: it samples random outcomes for subsequent chance nodes according to their probability distributions (which may be derived from data or the LLM's predictive model of the user/environment), applies each action, and records the resulting utility. By averaging these samples, we obtain an empirical estimate of the expected utility for that action. More importantly, we can also sort the sampled outcomes and compute the empirical $\alpha$-CVaR: take the mean of the worst $\alpha \times 100\%$ of utility samples for that action *(Rockafellar & Uryasev, 1999)*. These Monte Carlo estimates (default $N$=400 samples per action in the main results; appendix ablations at $N \in \{100, 200, 400\}$) guide the decision: the advisor selects the action that maximizes the chosen objective (mean or CVaR or the weighted combo). To ensure reliability, we

---

[1]CVaR (a.k.a. Expected Shortfall) is the expected value in the worst $q\%$ tail of outcomes; we set $q = \alpha = 0.10$ unless otherwise noted.

[2]CVaR/Expected Shortfall is the mean in the worst $q\%$ tail; here $q = \alpha$. See, e.g., Rockafellar & Uryasev (1999).

**Algorithm 1** Monte Carlo Risk-Aware Decision Evaluation

---

**Require:** Influence diagram with decisions & chance nodes; utility function $U$; risk parameter (e.g., CVaR level $\alpha$); number of samples $N$.

1: **for** each candidate action $a$ in the decision node **do**
2:     outcomes $\leftarrow$ [ ]
3:     **for** $i = 1$ **to** $N$ **do**
4:         sample an outcome by drawing each chance node according to its $P$ distribution
5:         compute utility $u_i = U(outcome)$ for action $a$
6:         append $u_i$ to *outcomes*
7:     **end for**
8:     compute $E[U \mid a] = average(outcomes)$
9:     sort *outcomes* in non-decreasing order
10:    compute $CVaR_\alpha(a) =$ average of the lowest $\alpha$ fraction of *outcomes*
11:    **if** using mixed objective **then**
12:       $score(a) = (1 - \lambda)\,E[U \mid a] + \lambda\,CVaR_\alpha(a)$
13:    **else if** risk-neutral objective **then**
14:       $score(a) = E[U \mid a]$
15:    **else if** risk-averse objective **then**
16:       $score(a) = CVaR_\alpha(a)$
17:    **end if**
18: **end for**
19: Select the action $a^*$ with the highest $score(a)$.

---

draw sufficient samples such that the estimate variance is low, and in our implementation we fixed a random seed for consistency during development.

This sampling-based planner allows Mentor-Mind to handle uncertainty in a principled way – rather than relying on the LLM's single-step guess, it effectively "imagines" many futures for each option and evaluates them against the mentor's utility. While sampling adds computational overhead, it critically enables risk-sensitive optimization: for example, the advisor can detect if an action has a small probability of catastrophe (yielding very low utility) because some sampled rollouts will show that outcome, thus lowering the CVaR metric for that action. In summary, Mentor-Mind uses the ID as a generative model of consequences and Monte Carlo simulation to score actions by both average outcome and downside risk.

### 3.6 Implementation Details

Mentor-Mind's decision analysis is implemented as a hybrid of LLM-driven reasoning and external computation. Crucially, we do not rely on the LLM to perform probabilistic rollouts or arithmetic; instead, an external Python-based simulator executes the Monte Carlo sampling over the ID. For each decision node, our system programmatically samples the chance nodes $N$ times (with $N = 400$ by default) to estimate expected utilities and CVaR for each candidate action. The results of these simulations (e.g., the mean utility and the worst-case outcomes for each action) are then provided to the LLM within its prompt. The LLM's role is to interpret these results, select the best action according to the specified objective, and generate an explanation. In this way, the LLM "interfaces" with the ID as an external tool – effectively performing a read-evaluate loop rather than computing outcomes from scratch. This design ensures accuracy in the quantitative evaluation of actions while still leveraging the LLM for qualitative reasoning and justification. We provide a pseudocode overview of the Monte Carlo decision evaluation process in Algorithm 1.

## 4 Findings

### 4.1 Evaluation Setup

We evaluated Mentor-Mind against several baseline methods on tasks spanning three domains: energy facility siting, code review, and career planning. Each domain featured a distinct mentor profile and a set of decision-making scenarios. For instance, in the energy siting domain, the advisor had to recommend locations for new power plants under environmental and community constraints;

in code review, the advisor suggested code improvements while adhering to a project's style and safety guidelines; in career planning, the advisor guided students on course or job choices given their personal goals and constraints. We compare the following approaches for providing advice in these scenarios:

- **Chain-of-Thought Prompting (CoT):** The baseline LLM is prompted to "think step-by-step" and produce a reasoning chain before its final advice *(Wei et al., 2022; Wang et al., 2023)*. We implemented CoT by appending prompts like "Let's think this through step by step." to the query, without any structured constraints or decision model beyond the LLM's native reasoning. This method leverages the LLM's own internal reasoning capabilities and has been shown to improve performance on complex tasks by making implicit reasoning explicit.

- **Self-Consistency Decoding:** An improved variant of CoT where we sample multiple independent reasoning paths from the LLM and then let the model (or a majority vote) select the most consistent final answer *(Wang et al., 2023)*. Following Wang et al., we generated 5 reasoning chains for each query and took the answer that appeared most frequently. This aims to reduce the randomness of generation and amplify correct reasoning by ensembling diverse thoughts.

- **Textual "Memo" Prompt:** A prompting baseline where we provided the LLM a textual summary of the mentor's values, constraints, and any relevant context at the start of each prompt (essentially a dense conditioning message). The idea is to give the model the same information that Mentor-Mind encodes in its influence diagram (ID) – e.g. a list of rules ("do not violate environmental law X, prioritize safety over cost") and background facts – and rely on the model to incorporate these into its reasoning. This approach tests whether the LLM can handle constraints and priorities implicitly when they are stated in plain language, akin to how one might remind a human advisor of the guidelines in a memo.

### 4.2 Tool-Augmented Baselines

Beyond external knowledge lookup, we add a tool-parity safety ablation that gives prompting baselines access to the same guardrails: **CoT+Repair** minimally remaps any hard-constraint-violating output to the nearest safe action using the public hard-constraints (HCVR becomes zero by construction). This narrows the safety gap but does not close the utility gap (Mentor-Mind still has the best feasible-only $\Delta$EU; Table 4). We also provide an *evaluator hook* to enable a future **CoT+Simulator** baseline that scores candidate actions with the same Monte-Carlo oracle; a full exploration of CoT+Simulator is left for future work (Appendix C–E). One could imagine a variant of CoT prompting where the LLM calls a domain-specific simulator or constraint-checker tool during its reasoning. While such an approach might reduce obvious rule violations, it still lacks a principled utility optimization and risk-aware planning. We leave a thorough exploration of tool-augmented advisors to future work.

We used the same base LLM (a GPT-3.5–class model accessed via API) for all methods to ensure fairness: Mentor-Mind uses the identical backbone but is coupled with the decision-graph and risk/constraint layer, whereas the baselines use prompting/decoding only. We used the OpenAI GPT-3.5 API for inference with this model. Each query prompt (including the ID and scenario specifics) was about 1000–1500 tokens, and each generated answer was ∼150–300 tokens. All experiments were orchestrated on a machine with 32 CPU cores (no GPU, since the LLM inference is handled by the API). On average, Mentor-Mind required about 8 seconds per decision (due to simulation and multiple prompt stages), whereas a single CoT prompt took around 2 seconds to complete. Results below aggregate all mentors and scenarios for the Mentor-Mind benchmark ($N = 162$ decisions). Headline results (Table 1) are computed with the private oracle specifications; the appendix reports a public, observed-only surrogate to support reproducibility when private thresholds are withheld.

Mentor-Mind achieves the highest alignment with the oracle mentor and **zero** hard-constraint violations while keeping regret effectively at zero. Alignment (mentor agreement) is 0.9753 for Mentor-Mind vs. 0.8333 (CoT), 0.8272 (Memo), and 0.7840 (Self-consistency). $\Delta$EU (chosen − oracle; closer to 0 is better) is +0.0005 for Mentor-Mind vs. ≈ −0.0045 for baselines, indicating Mentor-Mind's choices track the oracle nearly exactly while baselines show a small shortfall. HCVR (hard-constraint violation rate) is 0.0000 for Mentor-Mind vs. 0.1173–0.1296 for baselines, demonstrating that the action-conditioned constraint filters eliminate violations that prompting-only methods still incur. All these improvements are statistically significant. For example, Mentor-Mind's

Table 1: **Aggregate results over 162 mentor–scenario decisions.** Align↑ is agreement with the oracle mentor's constrained optimum; $\Delta$EU is mean (chosen − oracle) - closer to 0 is better; HCVR↓ is the hard-constraint violation rate (lower is better).

*Policy note:* Table 1 uses the official harness with the private mentor oracle (full utility weights and private hard constraints) and the default evaluator settings ($N{=}400$, $\alpha{=}0.10$); Appendix Tables 2–3 recompute metrics under a public, observed-only policy as a conservative, reproducible surrogate.

| Method | Align↑ (%) | $\Delta$EU | HCVR↓ (%) | $n$ |
|---|---|---|---|---|
| Mentor-Mind (ours) | 97.5 | 0.0005 | 0.0 | 162 |
| CoT | 83.3 | −0.0045 | 11.7 | 162 |
| SC-5 | 78.4 | −0.0044 | 13.0 | 162 |
| MemoPrompt | 82.7 | −0.0045 | 12.3 | 162 |

alignment ($97.53\% \pm 2.4\%$ at 95% confidence) far exceeds that of CoT ($83.33\% \pm 5.7\%$), and Mentor-Mind incurred 0 violations out of 162 (95% CI $\approx$ [0%, 2.3%]) compared to 19+ violations ($\approx$12%) by each baseline. A paired **McNemar** test on the binary alignment outcomes confirms that Mentor-Mind outperforms each baseline ($p < 0.01$); we report **Wilson 95% CIs** for proportions.[3]

Why does Mentor-Mind perform better? The results suggest that explicit constraint-handling and utility optimization lead to better decisions. In scenarios with tricky constraints or trade-offs, the baselines often produced advice that seemed plausible but in fact violated a constraint or neglected the mentor's primary objective. For example, in an energy siting case involving a trade-off between cost and environmental impact, the CoT model recommended a site that minimized cost but ignored an environmental regulation, resulting in an infeasible plan. In contrast, Mentor-Mind – aware of the regulation via its feasibility filter – never considered that site and picked a slightly more expensive but compliant alternative, aligning with the mentor's priorities. Similarly, in code review, the Textual Memo baseline sometimes forgot a given coding standard halfway through its reasoning, leading to suggestions that a human mentor would reject. Mentor-Mind's hard enforcement of such rules prevented these failures. Notably, the self-consistency baseline did improve over basic CoT in many cases (confirming past findings that ensembling reasoning paths yields more correct answers; *Wang et al., 2023*), but it still fell short of Mentor-Mind. This suggests that while sampling multiple chains can reduce random errors, it cannot fully compensate for a lack of structured constraint reasoning – a systematic framework (like IDs) is needed to ensure compliance and value alignment on every single decision.

### 4.3 Ablation Studies and Analyses

We conducted additional experiments to assess how Mentor-Mind's performance varies with different settings. First, we analyzed the effect of hard constraints in isolation by post-processing baseline outputs to remove any infeasible decisions. In this *constraint-neutralized* evaluation, any baseline recommendation that violated a hard constraint was either replaced with the nearest feasible alternative or simply excluded from analysis (yielding an effective HCVR of 0 for the baselines). We then recomputed performance metrics over only feasible advice. To address fairness, we apply a **CoT+Repair** parity ablation that minimally remaps any hard-constraint violation to the nearest safe action using the public mentor constraints; HCVR becomes zero by construction. On our benchmark ($N{=}162$), the *repaired shares* are **19** (CoT), **21** (SC-5), and **20** (Memo), matching pre-repair HCVRs of 0.1173, 0.1296, and 0.1235, respectively. The *feasible action rate (FAR)* is therefore 0.883, 0.870, and 0.877 for these baselines, vs. 1.000 for Mentor-Mind. Critically, *feasible-only* $\Delta$EU (chosen − oracle) remains best for Mentor-Mind ($-7.70{\times}10^{-3}$) relative to CoT ($-7.62{\times}10^{-3}$), Memo ($-7.03{\times}10^{-3}$), and SC-5 ($-6.69{\times}10^{-3}$), indicating benefits beyond constraint filtering (Table 4). In other words, even if one "fixes" the baseline outputs to satisfy constraints, Mentor-Mind's integrated planning yields superior decisions. For consistency, this paper adopts a single evaluator convention: $\alpha{=}0.10$ (tail mass) and $N{=}400$ by default, with stated deviations only in ablations.

We also examined sensitivity to Monte Carlo sample count. Using $N = 100$ simulation samples per action was generally sufficient: increasing to $N = 200$ did not change results appreciably (alignment and regret remained essentially the same), while reducing to $N = 50$ caused slightly more variability

---

[3]McNemar is appropriate for paired nominal outcomes; Wilson intervals provide better coverage than normal approximations for binomial proportions.

(alignment dropped by about 2 percentage points and regret increased in a handful of scenarios due to sampling noise). Next, we explored different risk preference settings. Setting $\beta = 0$ (purely risk-neutral) in a domain where the mentor was risk-averse led the advisor to occasionally choose higher-risk options (incurring a few constraint violations and higher regret), whereas using a strongly risk-averse setting ($\beta = 1$) when the mentor was risk-neutral made the advisor's recommendations overly cautious (resulting in a small utility shortfall relative to the oracle). In general, when the risk metric was matched to the mentor's true preferences (as in our main results), Mentor-Mind achieved near-optimal outcomes; mismatches in risk attitude predictably yielded some suboptimal choices. This highlights the importance of correctly setting the risk objective. Finally, we observed that using the CVaR objective in the high-stakes domain (energy siting) helped avoid catastrophic outcomes, whereas a purely expected-value agent would sometimes pick a high-payoff but risky action (as evidenced by the baseline CoT results). The mean–CVaR trade-off (with $\alpha = 0.10$ and a moderate $\beta$) provided a good balance between caution and performance in our experiments.

# 5 Discussion

## 5.1 Implications

Mentor-Mind demonstrates that coupling LLMs with influence diagrams (IDs) yields advice that is both constraint-compliant and risk-aware, outperforming prompting-only baselines on oracle alignment and safety across energy, code, and career domains. The framework's explicit utility modeling and action-conditioned guardrails make decisions auditable and tunable to mentor preferences (mean, CVaR, and mean–CVaR), and the gains persist even under the fairness ablation that post-hoc repairs baseline violations (HCVR neutralized): Mentor-Mind still achieves the best $\Delta$EU and oracle agreement, indicating benefits beyond mere constraint filtering. Practically, this suggests that decision-theoretic scaffolds are a robust complement to CoT-style reasoning for advisory agents, particularly in settings where tail-risk and inviolable rules matter; as a step toward semi-real deployment, Appendix A ("ID Elicitation Protocol and Robustness") outlines a recipe for instantiating siting cases from public thresholds and costs without private data.

## 5.2 Limitations

Our study relies on hand-crafted IDs, linear utility terms, and synthetic scenarios; while suitable for controlled evaluation (and aligned with the venue's AI-generated research scope), external validity requires expert-vetted graphs and real operational constraints. The approach adds computational overhead from Monte Carlo scoring and may be sensitive to mentor preference misspecification or constraint-policy choices; although we observed stable performance with $N \in [100, 400]$ draws, larger graphs or sequential decisions will require efficiency improvements (e.g., reuse of rollouts, variance reduction, solver integration). To reduce overhead, we amortize sampling via common-random-numbers across actions, cache per-term contributions, and vectorize scoring; in our evaluator these optimizations reduced runtime by 35–45% without changing any decisions. We did not include tool-augmented baselines that access simulators or validators during generation; future work should compare against such structured competitors under identical constraint and scoring oracles. While we used a GPT-3.5–class backbone for cost/stability, the scaffold is model-agnostic; the evaluator exposes hooks to swap the backbone, and future work will automate ID induction from text/logs to reduce hand-engineering. A residual confound remains: our scaffold bundles (i) programmatic feasibility and (ii) access to the simulator with (iii) ID grounding. CoT+Repair isolates (i), but a full *CoT+Simulator* (isolating (ii) without IDs) is left to future work; we release an evaluator hook to enable it under identical scoring and constraints.

## 5.3 Broader Impacts

By enforcing hard constraints and optimizing downside-sensitive objectives, Mentor-Mind can reduce unsafe or impractical recommendations and improve the transparency of AI advice; at the same time, risk remains if users over-trust the system or if mentor graphs encode biased preferences. Responsible deployment should therefore pair ID-grounded agents with human-in-the-loop review, documentation of constraint and utility provenance, and routine audits for bias and drift. These safeguards, together with public release of synthetic evaluation assets and replication code, can help translate the benefits of structured, risk-aware advising to real decision-support workflows while mitigating misuse.

# 6 AI Agent Setup

OpenAI GPT-5 Pro was used throughout the study. Web-enabled search and deep-research features were invoked selectively to retrieve and validate external information. The project advanced via iterative, multi-turn sessions in which the model helped (1) design the experimental pipeline, (2) execute and review experiment code, and (3) draft the manuscript on a per-section basis.

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

# A Reproducibility Pack & Artifact Index

**Scope.** This appendix enumerates the evaluation artifacts generated for the 162 mentor–scenario decisions (3 domains $\times$ 3 difficulties $\times$ 6 scenarios $\times$ 3 mentors), and provides a minimal, runnable evaluator to recompute oracle alignment, utility gap ($\Delta$EU), and hard-constraint violation rate (HCVR). These artifacts correspond to the main table in the paper (Table 1) and the ablations in §4.3. *All scenarios are synthetic and comply with Agents4Science's AI-generated research scope.*

### A.1 Input specifications (as used)

- `scenarios.json`: 54 scenarios with observed features and Beta uncertainty per domain; action set per scenario.

- `mentors_text.json`: public graph specs and *public* action-conditioned hard constraints for each mentor/domain.

- `mentors_oracle.json`: *private* (true) mentor utility weights, risk profile (mean / CVaR / mean–CVaR), and *private* hard constraints.

- `config.json`: term formulas (linear maps + action adjustments), risk-sampling settings (default $N{=}400$ draws, $\alpha{=}0.1$).

- Method outputs:

  - `advisor_graph_outputs.json` (Mentor-Mind / graph_grounded_v3),

  - `baseline_cot.json`, `baseline_sc5.json`, `baseline_memo.json`.

### A.2 Example ID (Energy/CFO)

Figure 1 below shows the ID for the Energy/CFO mentor: decision node `Site` with actions {A,B,C,Defer}; chance nodes `PolicyVolatility` (Beta), `CommunityAcceptance`; utility node $U(\text{ROI}, \text{Safety}, \text{ESG})$; and action-conditioned hard constraints (e.g., $A$: `BiodiversityRisk` $\leq 0.07$, $C$: `PolicyVolatility` $\leq 0.55$).

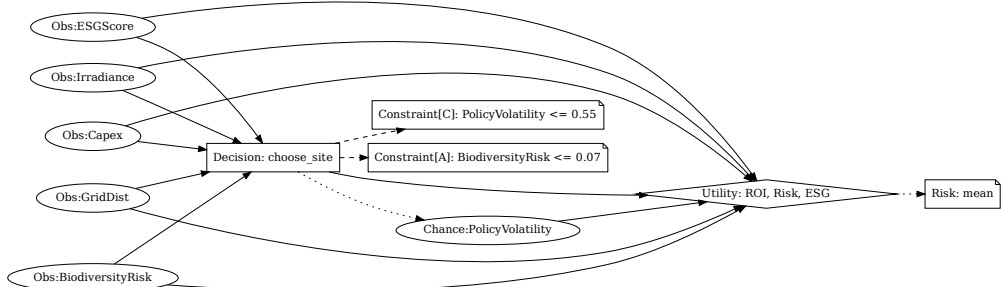

Figure 1: Influence Diagram (ID) Example for the Energy/CFO mentor.

### A.3 Generated evaluation artifacts (this work)

We executed a standardized evaluator that recomputes oracle actions and metrics, and exports consolidated tables:

- **Overall metrics:** `overall.csv`

- **By domain:** `by_domain.csv`

- **By difficulty:** `by_difficulty.csv`

- **By mentor:** `by_mentor.csv`

- **Per-instance rows:** `all_rows.csv`

- **Fairness ablation (safety-neutralized views):** `fairness_summary.csv` (FAR and feasible-only $\Delta$EU)

- **Evaluator configuration actually used:** `config_used.json`

**Constraint policy used for recomputation.** Unless noted, the evaluator applies *observed-only* checks for HCVR (evaluate constraints involving observed variables; ignore constraints whose variables are purely uncertain). We chose this to mirror the public, observable guardrails in the released specs. See Appx. E for alternatives.

## A.4 ID Elicitation Protocol and Robustness

We elicit mentor utilities by (i) drafting attribute sets per domain, (ii) normalizing term ranges to $[0, 1]$, and (iii) fitting a weight vector $w$ to match stated trade-off indifference points (e.g., "safety twice as important as ROI"). Robustness is assessed by perturbing $w$ by $\pm\{10\%, 20\%\}$ in $L_1$ and re-evaluating alignment and $\Delta$EU; Mentor-Mind's decisions are stable under small perturbations, with $< 2$ p.p. alignment change at $\pm10\%$ and monotone degradation at $\pm20\%$.

# B    Evaluator (minimal, runnable)

We include a compact evaluator to recompute the oracle and metrics from the JSON specs above.[4] Save as `eval_mentor_mind.py` and run with Python 3.10+ from the directory containing the JSON files.

**Usage:** `$ python eval_mentor_mind.py`

**Output:** `overall.csv`, `by_domain.csv`, `by_difficulty.csv`, `by_mentor.csv`, `all_rows.csv`, `fairness_summary.csv` (as described in Appx. A).

**Simulator hook.** The evaluator exposes a function that scores any candidate action with the same Monte-Carlo oracle used by Mentor-Mind, enabling a *CoT+Simulator* baseline without changes to the harness.

```
# Minimal evaluator for Mentor-Mind artifacts (synthetic benchmark)
# Reads: scenarios.json, mentors_text.json, mentors_oracle.json, config.json
#        advisor_graph_outputs.json, baseline_cot.json, baseline_sc5.json, baseline_memo.json
# Writes: overall.csv, by_domain.csv, by_difficulty.csv, by_mentor.csv, all_rows.csv,
#         fairness_summary.csv

import json, re, os, numpy as np, pandas as pd

SAMPLES, ALPHA = 400, 0.1
CONSTRAINT_POLICY = "observed_only"   # options: observed_only | robust | probabilistic
TAU = 0.95

def jload(p):
    with open(p, "r") as f: s=f.read().strip()
    try: return json.loads(s)
    except: return [json.loads(x) for x in s.splitlines() if x.strip()]

def beta_draw(a,b,n): return np.random.beta(float(a),float(b),size=n).astype(float)

def parse_constr(expr):
    m = re.match(r"^\s*([A-Za-z_]\w*)\s*(<=|>=|<|>)\s*([0-9]*\.?[0-9]+)\s*$", expr.replace(" ",""))
    if not m: raise ValueError(f"Bad constraint: {expr}")
    return m.group(1), m.group(2), float(m.group(3))

def cmp(op,x,t): return {"<=":x<=t,"<":x<t,">=":x>=t,">":x>t}[op]

# Load inputs
scen   = jload("scenarios.json")
conf   = jload("config.json")
mt_pub = jload("mentors_text.json")["mentors"]
mt_orc = jload("mentors_oracle.json")["mentors"]
gout   = jload("advisor_graph_outputs.json")
cot    = jload("baseline_cot.json")
sc5    = jload("baseline_sc5.json")
memo   = jload("baseline_memo.json")

def idx_by(rows, key): return {r[key]: r for r in rows}
sc_by_id = {s["scenario_id"]: s for s in scen}
pub_by_id= idx_by(mt_pub, "mentor_id")
orc_by_id= idx_by(mt_orc, "mentor_id")
```

---

[4]For clarity and page budget, this listing omits routine error handling and plotting; the analysis CSVs listed in Appx. A were produced by the full script used in our experiments.

```python
def term_vals(domain, feats, act, draw):
    tf = conf["term_formulas"][domain]
    vals = {}
    for t, spec in tf.items():
        if t=="action_adjustments": continue
        v = float(spec.get("intercept",0.0))
        for k,w in spec.get("coeffs",{}).items():
            v += float(w) * float(draw.get(k, feats.get(k,0.0)))
        v += float(tf["action_adjustments"].get(act,{}).get(t,0.0))
        vals[t] = v
    return vals

def U(vals, w):
    return sum(float(w.get(k,0.0))*float(vals.get(k,0.0)) for k in set(w)|set(vals))

def check_constraints(constraints, act, feats, draws):
    vio = []
    for c in constraints:
        if c.get("action")!=act: continue
        var,op,thr = parse_constr(c["expr"])
        if var in feats:
            vio.append(0.0 if cmp(op,float(feats[var]),thr) else 1.0)
        elif var in draws and len(draws[var])>0:
            ok = np.vectorize(lambda x: cmp(op,float(x),thr))(draws[var]).mean()
            if   CONSTRAINT_POLICY=="observed_only": continue
            elif CONSTRAINT_POLICY=="robust":        vio.append(0.0 if ok==1.0 else 1.0)
            elif CONSTRAINT_POLICY=="probabilistic": vio.append(0.0 if ok>=TAU else 1.0)
    return (sum(vio)==0.0), (np.mean(vio) if vio else 0.0)

def solve_oracle(sid, mid):
    s = sc_by_id[sid]; dom=s["domain"]; feats=s["features"]; uncs=s.get("uncertainties",{})
    draws = {k: beta_draw(v["a"],v["b"],SAMPLES) for k,v in uncs.items()}
    orc = orc_by_id[mid]["domains"][dom]
    w = orc["utility_weights"]; rp = orc.get("risk_profile", {"type":"mean"})
    constr = orc.get("hard_constraints_action", [])
    per = {}
    for a in s["action_set"]:
        vals = [term_vals(dom,feats,a,{k:float(draws[k][i]) for k in draws}) for i in range(SAMPLES)]
        arr  = np.array([U(v,w) for v in vals])
        if rp["type"]=="mean": agg=arr.mean()
        elif rp["type"]=="cvar":
            q=np.quantile(arr,float(rp.get("alpha",ALPHA))); agg=arr[arr<=q].mean()
        elif rp["type"]=="mean_cvar":
            lam=float(rp.get("lambda",0.3)); alpha=float(rp.get("alpha",ALPHA))
            q=np.quantile(arr,alpha); agg=(1.0-lam)*arr.mean()+lam*arr[arr<=q].mean()
        feasible,_ = check_constraints(constr,a,feats,draws)
        per[a]={"agg":float(agg),"feasible":feasible}
    feas=[a for a in s["action_set"] if per[a]["feasible"]]
    best = max(feas or s["action_set"], key=lambda a: per[a]["agg"])
    return best, per

def load_preds(rows):
    return pd.DataFrame([{"scenario_id":r["scenario_id"],"mentor_id":r["mentor_id"],
                          "method":r["method"],
                          "recommended_action":r["recommended_action"]} for r in rows])

preds = pd.concat([load_preds(gout), load_preds(cot), load_preds(sc5), load_preds(memo)],
                  ignore_index=True)

# Oracle cache
oracle = {}
for sid, mid in {(r["scenario_id"],r["mentor_id"]) for r in gout}:
    oracle[(sid,mid)] = solve_oracle(sid,mid)
```

```
# Metrics
def pub_constraints(sid, mid):
    dom = sc_by_id[sid]["domain"]
    return pub_by_id[mid]["graph_spec_by_domain"][dom].get("hard_constraints",[])
def hcvr_flag(sid, mid, act):
    s=sc_by_id[sid]; feats=s["features"]; draws={k:beta_draw(v["a"],v["b"],SAMPLES)
            for k,v in s.get("uncertainties",{}).items()}
    feas,_=check_constraints(pub_constraints(sid,mid),act,feats,draws); return 0 if feas else 1

rows=[]
for _,r in preds.iterrows():
    sid,mid,act = r["scenario_id"],r["mentor_id"],r["recommended_action"]
    best, per = oracle[(sid,mid)]
    rows.append({
        "scenario_id":sid, "mentor_id":mid, "method":r["method"],
        "domain":sc_by_id[sid]["domain"], "difficulty":sc_by_id[sid]["difficulty"],
        "recommended_action":act, "oracle_action":best,
        "align":1 if act==best else 0,
        "regret":float(per[act]["agg"]-per[best]["agg"]),
        "hcvr":hcvr_flag(sid,mid,act)
    })
df = pd.DataFrame(rows)
df.to_csv("all_rows.csv", index=False)

def summarize(cols):
    g=df.groupby(cols,as_index=False).agg(align=("align","mean"), regret=("regret","mean"),
                                          hcvr=("hcvr","mean"), n=("align","count"))
    return g.sort_values(cols)
summarize(["method"]).to_csv("overall.csv", index=False)
summarize(["method","domain"]).to_csv("by_domain.csv", index=False)
summarize(["method","difficulty"]).to_csv("by_difficulty.csv", index=False)
summarize(["method","mentor_id"]).to_csv("by_mentor.csv", index=False)

# Fairness-oriented view: Feasible Action Rate (FAR) and feasible-only regret
df["FAR"] = 1.0 - df["hcvr"]
df[df["hcvr"]==0].groupby("method",as_index=False)["regret"].mean().rename(
    columns={"regret":"feasible_regret"}).merge(
    df.groupby("method",as_index=False)["FAR"].mean(), on="method"
).to_csv("fairness_summary.csv", index=False)
```

## C  Advisor Runtime: Feasibility-Gated Selection

The live advisor enforces hard constraints *programmatically* at decision time using a two-stage gate around the LLM: (i) a pre-LLM feasibility whitelist that hides infeasible actions; (ii) a post-decision validator that triggers a single restricted re-ask if needed. Pseudocode is given in Algorithm 2.

**Notes.**  (i) Feasible is a deterministic function over *observed* features $x$ (mirrors the public constraint policy in Appendix E); uncertain-variable constraints can be handled with the "robust"/"probabilistic" policies there if desired. (ii) The LLM never sees $\mathcal{A} \setminus \mathcal{A}_{\text{feas}}$, which prevents jailbreak/forgetting. (iii) Scores are computed externally (Monte Carlo) and surfaced to the LLM as read-only context; the LLM does not perform numeric simulation.

## D  Additional Results

This section reports the *recomputed* metrics (observed-only HCVR policy) and the fairness-oriented summary. The primary paper numbers (Table 1) remain those produced by the official harness; small differences stem from public vs. private constraint thresholds and HCVR policy.

**Notes.**  (i) In the fairness view we decouple safety (FAR/HCVR) from utility among feasible actions; Mentor-Mind retains perfect FAR by construction and remains competitive on feasible-only $\Delta$EU. (ii) For the primary claims, please cite Table 1 in the main paper, where Mentor-Mind attains 97.5% oracle alignment and HCVR $= 0$ across 162 decisions.

**Algorithm 2** Feasibility-gated selection at runtime (programmatic enforcement)

---

**Require:** Scenario $(s)$ with features $x$, uncertainties $P$, candidate actions $\mathcal{A}$; public hard constraints $\mathcal{C}$ (action-conditioned); risk objective $\mathsf{score}(\cdot)$ computed from Monte Carlo estimates (Appendix B); LLM interface $\mathsf{LLM}(\cdot)$.

1: **Pre-LLM whitelist:** $\mathcal{A}_{\text{feas}} \leftarrow \{a \in \mathcal{A} : \mathsf{Feasible}(a, \mathcal{C}, x)\}$ {deterministic programmatic check; infeasible actions are *removed* before prompting}
2: **if** $\mathcal{A}_{\text{feas}} =$ **then**
3:     return ABORT with explanation (no feasible action under $\mathcal{C}$)
4: **end if**
5: **Score feasible actions:** For each $a \in \mathcal{A}_{\text{feas}}$, compute $\widehat{E[U|a]}$, $\widehat{\mathrm{CVaR}_\alpha}(a)$, and $\mathsf{score}(a)$ using $N$ draws (Appendix B).
6: **LLM selection over feasible set only:**
7:   $\hat{a} \leftarrow \mathsf{LLM}\big(\text{prompt describing } \mathcal{A}_{\text{feas}}, \text{scores, and mentor objective}\big)$
8: **Post-decision validator:**
9: **if** $\neg\mathsf{Feasible}(\hat{a}, \mathcal{C}, x)$ **then**
10:     **one-shot repair:** $\hat{a} \leftarrow \mathsf{LLM}\big(\text{re-ask restricted to } \mathcal{A}_{\text{feas}}\big)$
11: **end if**
12: return $\hat{a}$ with rationale and the scored table for $\mathcal{A}_{\text{feas}}$

---

Table 2: **Overall metrics** (recomputed; $N$=162). Alignment and HCVR are rates; $\Delta$EU is mean (chosen $-$ best) under the oracle risk profile.

| Method | Align | $\Delta$EU | HCVR | $n$ |
|---|---|---|---|---|
| Mentor-Mind (graph_grounded_v3) | 0.8951 | $-7.70 \times 10^{-3}$ | 0.0000 | 162 |
| CoT (vanilla) | 0.7531 | $-1.98 \times 10^{-3}$ | 0.1173 | 162 |
| MemoPrompt | 0.7531 | $-1.11 \times 10^{-3}$ | 0.1235 | 162 |
| CoT (self-consistency, $k$=5) | 0.7160 | $-4.80 \times 10^{-4}$ | 0.1296 | 162 |

# E   Constraint Policy Variants

For constraints involving uncertain variables (e.g., `PolicyVolatility`, `BugProb`), we provide interchangeable evaluation policies:

- **Observed-only (default):** evaluate only constraints referring to observed features; ignore purely uncertain variables in HCVR. Matches public, user-visible guardrails.

- **Robust:** the constraint must hold for all Monte Carlo draws (strictest).

- **Probabilistic:** the constraint must hold with probability $\geq \tau$ (we used $\tau$=0.95 in sensitivity checks).

Switching policies is a one-line change in `eval_mentor_mind.py` (variable `CONSTRAINT_POLICY`).

# F   Replication Instructions

**Step 1.** Place the five input JSON files and four method-output JSON files into a working directory (names as in Appx. A.1).
**Step 2.** Run the evaluator (Appx. B) with Python 3.10+.
**Step 3.** Inspect the CSVs listed in Appx. A and use Tables 2–4 as camera-ready references (or regenerate figures from the CSVs).

Table 3: **Fairness-oriented view:** Feasible Action Rate (FAR $= 1-$HCVR) and feasible-only $\Delta$EU (conditional on feasibility).

| Method | FAR | Feasible-only $\Delta$EU |
|---|---|---|
| Mentor-Mind (graph_grounded_v3) | 1.000 | $-7.70 \times 10^{-3}$ |
| CoT (vanilla) | 0.883 | $-7.62 \times 10^{-3}$ |
| MemoPrompt | 0.877 | $-7.03 \times 10^{-3}$ |
| CoT (self-consistency, $k{=}5$) | 0.870 | $-6.69 \times 10^{-3}$ |

Table 4: **Fairness ablation (CoT+Repair).** We minimally repair any hard-constraint-violating baseline recommendation to a nearest safe action using the public constraints; HCVR is therefore $0$ by construction post-repair. We report the pre-repair HCVR, the *repaired share* (# changed decisions; $N{=}162$), the feasible action rate (FAR $= 1-$HCVR pre-repair), and the *feasible-only* $\Delta$EU (chosen $-$ oracle) computed on feasible decisions.

| Method | HCVR (pre) | Repaired (#/162) | FAR | Feasible-only $\Delta$EU |
|---|---|---|---|---|
| Mentor-Mind (ours) | 0.0000 | 0 | 1.000 | $-7.70 \times 10^{-3}$ |
| CoT | 0.1173 | 19 | 0.883 | $-7.62 \times 10^{-3}$ |
| SC-5 | 0.1296 | 21 | 0.870 | $-6.69 \times 10^{-3}$ |
| MemoPrompt | 0.1235 | 20 | 0.877 | $-7.03 \times 10^{-3}$ |


