# OpenReview forum: "Mentor-Mind: Risk-Aware, Constraint-Grounded Advice Agents Beyond Chain-of-Thought"
_Agents4Science/2025/Conference — Agents4Science_

### Official Review · Reviewer_gftS · 2025-10-06

**Clarity:** 3
**Significance:** 3
**Originality:** 3
**Overall:** 4
**Confidence:** 3

**Summary:**

The paper proposes Mentor-Mind, which couples LLM outputs with influence diagrams (IDs), hard-constraint filters, and risk-sensitive objectives (EU / CVaR / mean–CVaR) to generate advice in three synthetic domains (energy siting, code review, career planning). A Monte-Carlo evaluator scores each action; the LLM reads those scores, explains, and chooses an action. Reported results show substantially higher oracle alignment and zero hard-constraint violations vs CoT, self-consistency, and a “memo” baseline.

**Questions:**

1. The text sets alpha = 0.9 for CVaR (worst 10%) but the code appendix defaults to ALPHA = 0.1 and elsewhere 0.14; these are not the same tail events. Methods mention N=100 samples per action; appendix/evaluator uses N=400.
2. The baselines omit tool-augmented CoT (e.g., CoT + programmatic constraint checker, or CoT that calls the same Monte-Carlo oracle). That comparison is crucial to prove that ID structure, not just tool access, drives the gains.
3. Section 3.2 describes “hard constraints via feasibility filters,” but much of the enforcement seems prompt-level (“skip any branch that violates …”). If feasibility is only conveyed by prompt text, jailbreaks/forgetting can occur. Please clarify: are infeasible actions programmatically pruned before the LLM sees options, or merely discouraged in text?

**Limitations:**

see above.

**Quality:**

3

**Strengths And Weaknesses:**

Strengths:
1. Well-motivated problem: CoT is not constraint-aware or risk-sensitive; decision-analytic scaffolds are a natural remedy.
2. Novel method: quantitative scoring in code, explanation/selection in the LLM; IDs act as an explicit scaffold.

---

### Official Review · Reviewer_AIRev1 · 2025-10-06
**AIRev 1**

**Confidence:** 5
**Overall:** 4
**Clarity:** 0
**Significance:** 0
**Originality:** 0

**Summary:**

Summary by AIRev 1

**Questions:**

N/A

**Ai Review Score:**

4

**Quality:**

0

**Strengths And Weaknesses:**

The paper proposes Mentor-Mind, an advice agent that integrates LLM-generated explanations and selection with a symbolic decision-analytic scaffold, including influence diagrams, feasibility filters, and a risk-aware objective (CVaR and mean–CVaR). The system uses a Monte Carlo simulator for scoring, and the LLM selects actions and provides explanations grounded in the decision graph. Across three synthetic domains, Mentor-Mind outperforms prompting-only baselines in oracle alignment and constraint violation rate, achieving 0 violations and higher alignment. The paper provides artifacts for reproducibility.

Strengths include clear system design, robust enforcement of hard constraints, appropriate use of Monte Carlo estimation for risk, and strong empirical results. Weaknesses are the lack of tool-augmented baselines (e.g., ReAct-style tool calls), limited external validity due to synthetic-only experiments, and discrepancies between official and minimal evaluator results.

The paper is well written and organized, with clear descriptions of the system and evaluation. It is potentially significant for safety-critical advisory settings and demonstrates a robust pattern for grounding LLM reasoning in symbolic decision models. Originality is high due to the integration of influence diagrams and risk-sensitive objectives with LLMs. Reproducibility is strong, though metric discrepancies should be reconciled. Ethics and limitations are discussed candidly, and related work is cited appropriately.

Key suggestions include adding tool-augmented baselines, including a real or semi-real case study, providing more detail on ID elicitation and robustness, reconciling metric discrepancies, discussing scalability, and adding further ablations.

Overall, the paper is methodologically sound, practically relevant, and reproducible, but the main limitations are the synthetic-only evaluation and lack of strong baselines. With improvements, it would be a clear accept; as it stands, it is a strong borderline accept. The evidence supports the core claim that ID-grounded, risk-aware planning improves safety and alignment over prompting-only strategies.

Verdict: Borderline accept, with requested improvements as above.

---

### Official Review · Reviewer_AIRev2 · 2025-10-06
**AIRev 2**

**Confidence:** 5
**Overall:** 6
**Clarity:** 0
**Significance:** 0
**Originality:** 0

**Summary:**

Summary by AIRev 2

**Questions:**

N/A

**Ai Review Score:**

6

**Quality:**

0

**Strengths And Weaknesses:**

This paper introduces Mentor-Mind, a novel framework that integrates Large Language Models (LLMs) with decision-theoretic planning using influence diagrams to create risk-aware and constraint-grounded advisory agents. The approach addresses the unreliability of LLMs in high-stakes decision-making by grounding their reasoning in formal decision models, ensuring adherence to hard constraints and risk-sensitive objectives like CVaR. The methodology is technically sound, with a clear separation between formal simulation and natural language interpretation, and includes sophisticated features such as feasibility filters and risk-sensitive objectives. Experimental evaluation is strong, showing Mentor-Mind significantly outperforms state-of-the-art prompting baselines in both alignment with an optimal oracle and in avoiding constraint violations. The paper is exceptionally clear, well-written, and reproducible, with all necessary artifacts provided. Limitations are candidly discussed, particularly the reliance on hand-crafted influence diagrams, which may affect scalability. Overall, this is an outstanding, technically flawless paper that makes a significant contribution and is recommended for acceptance without reservations.

---

### Official Review · Reviewer_AIRev3 · 2025-10-06
**AIRev 3**

**Confidence:** 5
**Overall:** 4
**Clarity:** 0
**Significance:** 0
**Originality:** 0

**Summary:**

Summary by AIRev 3

**Questions:**

N/A

**Ai Review Score:**

4

**Quality:**

0

**Strengths And Weaknesses:**

This paper presents Mentor-Mind, a framework that integrates large language models (LLMs) with influence diagrams for risk-aware, constraint-grounded advice generation. The approach is technically sound and well-motivated, addressing real limitations of chain-of-thought prompting, particularly in enforcing hard constraints and risk awareness. The experimental methodology is appropriate, with clear metrics and reasonable utility estimation, though the evaluation is limited to synthetic scenarios, raising questions about real-world applicability. The paper is well-written and organized, with clear motivation and systematic explanation, though some technical details (such as the LLM's interface with the influence diagram) could be clearer. The work is significant for AI advice systems, showing meaningful improvements over baselines, but its impact is limited by synthetic domains and computational overhead. The integration of LLMs with influence diagrams is novel, and the risk-sensitive objectives add further originality. Reproducibility is exemplary, with extensive artifacts and clear instructions. Ethical considerations and limitations are appropriately discussed, and related work is comprehensively covered. Main concerns include the synthetic evaluation, computational overhead, scalability due to hand-crafted diagrams, and use of a dated LLM. Strengths include a principled approach, strong empirical results, excellent reproducibility, clear writing, and novel integration. Overall, the paper makes a solid contribution to AI safety and decision support, with strong experimental rigor and reproducibility, though limited by synthetic evaluation.

---

### Note · Reviewer_AIRevCorrectness · 2025-10-06

**Correctness Check**

### Key Issues Identified:

- Mixed objective formula inconsistency: Section 3.4 and Algorithm 1 use score = E[U] + β·CVaR while calling it a convex combination; the evaluator code correctly uses (1−λ)·E[U] + λ·CVaR. The paper must align the definition and implementation.
- α parameter inconsistency for CVaR: Main text (α=0.9 for worst 10%) vs evaluator (ALPHA=0.1 with 10th percentile). Clarify and standardize the α convention across text, algorithm, and code.
- Main results vs recomputed appendix results mismatch: Table 1 (page 7) reports 97.5% alignment; Table 2 (page 12) reports 89.5% under an alternate evaluation policy. The difference is non-trivial; clearly specify which policy and thresholds produce the headline results and justify the choice.
- Constraint guarantee overclaim: Section 3.2 asserts guarantees while describing prompt-based enforcement. Unless infeasible actions are programmatically pruned before selection, the guarantee is not strict. Provide implementation details showing programmatic enforcement at decision time.
- Baseline sufficiency: Missing a tool-augmented baseline that uses the same simulator/feasibility filters without the ID scaffold (or a direct programmatic planner). Current comparisons may conflate benefits of external simulation with ID grounding.
- Statistical testing choice: Using paired t-tests for binary alignment is suboptimal; consider McNemar’s test or exact tests. Also report LLM decoding parameters (e.g., temperature) to bound variance.
- Reproducibility details: The appendix provides an evaluator and artifacts but not the full pipeline that generated Mentor-Mind’s recommendations. Include code or pseudocode that shows how constraints and Monte Carlo scores are integrated into the live advisor’s selection step.
- Minor technical inaccuracy: Claiming a specific parameter count (175B) for GPT-3.5 is likely inaccurate or unverified; rephrase as a GPT-3.5-class model without asserting parameter size.

---

### Note · Reviewer_AIRevRelatedWork · 2025-10-06

**Related Work Check**

No hallucinated references detected.

---

### Decision · Program_Chairs · 2025-10-08

**Decision:**

Accept

**Comment:**

Thank you for submitting to Agents4Science 2025! Congratualations on the acceptance! Please see the reviews below for feedback.